# Influencing Effects of Fabrication Errors on Performances of the Dielectric Metalens

**DOI:** 10.3390/mi13122098

**Published:** 2022-11-28

**Authors:** Guoqing Xu, Qianlong Kang, Xueqiang Fan, Guanghui Yang, Kai Guo, Zhongyi Guo

**Affiliations:** School of Computer and Information, Hefei University of Technology, Hefei 230009, China

**Keywords:** dielectric metasurfaces, metalens, fabrication errors, focusing efficiency

## Abstract

Despite continuous developments of manufacturing technology for micro-devices and nano-devices, fabrication errors still exist during the manufacturing process. To reduce manufacturing costs and save time, it is necessary to analyze the effects of fabrication errors on the performances of micro-/nano-devices, such as the dielectric metasurface-based metalens. Here, we mainly analyzed the influences of fabrication errors in dielectric metasurface-based metalens, including geometric size and shape of the unit element, on the focusing efficiency and the full width at half maximum (FWHM) values. Simulation results demonstrated that the performance of the metasurface was robust to fabrication errors within a certain range, which provides a theoretical guide for the concrete fabrication processes of dielectric metasurfaces.

## 1. Introduction

The traditional optical lens realizes imaging through the gradual accumulation of phase shift along the propagation path. However, it has some inherent shortcomings, such as bulk volume, complex shape and limited performance. Metasurface is a kind of planar array, composed of subwavelength resonant units arranged in periodic or aperiodic ways, providing an opportunity to artificially manipulate light [1,2,3]. Compared with the conventional optical lens, the optical lens based on metasurface has the advantages of being ultrathin, highly integrative, and versatile in function [4,5]. Due to metasurfaces possessing fascinating capabilities of light manipulation, they have been widely applied to various fields [6,7,8,9,10,11,12,13,14,15].

The metasurface can control the phase, amplitude, and polarization of the incident electromagnetic wave by changing the dimension parameters and orientations of the resonant units [16,17], so as to realize singular electromagnetic characteristics. At present, in miniaturized and integrated optical systems, metasurfaces have been designed to achieve polarization transformation [18], optical focusing [19,20,21], holographic imaging and filtering [22,23,24]. Typically, metasurface-based devices perform these functions, including flat, sub-wavelength size, and higher focusing efficiency [25,26,27,28,29], which are beneficial for the development of integrated devices.

At present, micro–nano fabrication technology [30] remains imperfect, though it is rapidly developing. Due to the subwavelength feature of unit cells, geometric errors are inevitably introduced in the fabrication process [31,32]. Ou et al. demonstrated a broadband achromatic metalens in the wavelength range of 3.5–5 μm with an all-silicon metasurface [33]. Their measured focusing efficiency was lower than the simulated, which could be attributed to fabrication errors. Jia et al. reported a transmissive focusing metalens in the terahertz range, based on an all dielectric metasurface that consisted of silicon nano-antennas [34]. Due to fabrication errors, the measured full width at half maximum (FWHM) at the focus was slightly larger than the theoretical result and focusing efficiency was lower than the simulated result.

In recent years, the influences of several common fabrication errors on the performances of metasurfaces have been investigated in the visible and near infrared ranges, where the metasurfaces are composited by unit structures with relatively low aspect ratio [35,36,37,38]. However, the unit cells of mid-infrared metalens have much higher aspect ratio, which might more easily introduce errors during fabrication. Obviously, the fabrication errors have an uncertain impact on performance parameters of the designed metalens. Therefore, it could be helpful to analyze the effect of fabrication errors on the performance of dielectric metasurface so as to reduce manufacturing costs and save time.

The manufacturing process of the metasurface, shown in Figure 1a, was as follows. Firstly, a GaAs film with a thickness of 3.6 μm was deposited on the GaF2 substrate by chemical vapor deposition. Secondly, the GaAs film was fabricated using electron beam lithography (EBL) technology. Finally, reactive ion etching with chlorine gas was performed in an inductively coupled plasma system. However, GaAs nano-bricks with high aspect ratio are easily damaged during the EBL process, inducing fabrication errors, such as geometric size and shape deviations of the GaAs nano-bricks [36], which affects the focusing efficiency and FWHM of the designed metalens.

In this paper, we numerically investigated the influences of different types of fabrication errors on the performances of focusing mesurfaces, such as the focusing efficiency and FWHM. The metasurface consisted of gallium arsenide (GaAs) nano-bricks and manipulated orthogonal linear polarizations simultaneously at a wavelength of 4.2 μm [39]. Three kinds of typical fabrication errors, including those affecting the geometric parameter and morphology of the metasurface, were systematically studied. The results could be helpful for the rapid development of metasurfaces from their fundamentals to applications.

## 2. Structure Design

The investigated metasurface was composed of an array of GaAs nano-bricks standing on a calcium fluoride (CaF_2_) substrate, as schematically shown in Figure 1a. The *x*- and *y*-polarized incident plane wave with wavelength of 4.2 μm could be focused at two different positions with the same focal length of 21 μm. The designed element is shown in Figure 1b with the height of the GaAs nano-bricks being *h* = 3.6 μm. The period was set as *P* = 1.8 μm. *L* and *w* denote the length and width of the GaAs nano-bricks, respectively, the values varying from 0.35 μm to 1.4 μm to achieve the gradient transmission phases covering 0–2π. Several typical fabrication errors of the metasurface could emerge during the manufacturing process [40,41], such as the following: (i) The height of the nano-bricks could be different from the designed value, especially for the metasurface with high aspect ratio; (ii) The top of the GaAs nano-bricks was an isosceles trapezoidal platform, as shown in Figure 1c, and *h* = *h*_1_ + *h*_2_. We kept *h*_1_ and *h*_2_ as constants and error was introduced by changing the inclination angle *θ* of the isosceles trapezoidal platform; (iii) The top of GaAs nano-bricks was a smooth elliptical hemisphere, as shown in Figure 1d, and the error was represented by the value of *h*_3_.

We performed numerical simulations and optimized our structure using a home-built program based on the FEM (finite element method). In the simulations, a plane wave with the wavelength of 4.2 μm propagating along the *z*-axis was incident from the substrate, and its polarization was along the *x*-axis or *y*-axis. To save simulation time and memory, the structure was assumed to be infinite in both *x* and *y* directions, and the periodic boundary conditions were adopted at the four sides of the simulated element. Meanwhile, to remove the simulated reflected waves from the top and bottom, PMLs (perfectly matched layers) were applied on the top and bottom boundaries in the *z* direction. In addition, to ensure simulation accuracy, the minimum mesh size was set to be 30 nm. The concrete optical constants of GaAs and CaF_2_ were taken from Ref. [42].

Before studying the fabrication errors, a double phase modulation metalens [43,44] was designed to focus *x*- and *y*-polarized incident plane waves at a wavelength of 4.2 μm. To this end, the metalens could be constructed by the elements of GaAs nano-bricks, whose phase distribution satisfied the following condition [42,43,44]:(1)$$\Phi \left(x,y\right)=\frac{2\pi }{\lambda }\left(\sqrt{{\left(x\pm x_{0}\right)}^{2}+{\left(y\pm y_{0}\right)}^{2}+f^{2}}-f\right)$$
where *λ* represents the incident wavelength, and *f* represents the predesigned focal length, *x* and *y* are the coordinate of the nano-bricks, *x*_0_ and *y*_0_ represent the offset distances of focus from the center of the metasurface along *x*- and *y*-axis, respectively. Herein, we assumed a two-dimensional metalens with 34 units, in which the nano-bricks were distributed along the *x*-axis. The corresponding phase profile of the focusing metalens could be written as $\Phi \left(x\right)=\frac{2\pi }{\lambda }\left(\sqrt{{\left(x\pm x_{0}\right)}^{2}+f^{2}}-f\right)$, and the focal length and the concrete offset distance were set as 21.0 μm and 14.4 μm, respectively. Once the phase profile was satisfied, the incident plane wave converged into a spherical wave.

The designed metalens consisted of elements of 34 units of GaAs nano-bricks, in which the structural parameters and corresponding phases of each nano-bricks are shown in Table 1.

Here, we introduce the designing of the metasurface based on GaAs nano-bricks with the help of double-phase modulation [45,46], and extending the modulation dimension of linearly polarized light. Due to the rectangular structure of the co-polarized light and the small polarization conversion of the non-rotating period nano-brick, the transmission phase modulation of the co-polarized light could easily be realized. As shown in Figure 2a,b, when the normal incident wavelength of *x*-polarized light was 4.2 μm, the phase and transmittance of transmitted *x*-polarized light could be expressed as a function of the length (*L*) and width (*w*) of the rectangular nano-brick in the range of 0.5~1.5 μm, respectively.

The phase modulation of transmitted light almost covered the whole 0~2π range, and higher transmittance could also be selected. Similarly, Figure 2c,d, show the concrete manipulating of phase and transmittance of transmitted *y*-polarized light, respectively. As can be seen from Figure 2, for any modulation phase of transmitted *x*- and *y*-polarization light, there existed a rectangular GaAs nano-brick, which could satisfy the needs of phase modulations of two orthogonal polarizations simultaneously, and the higher transmittance. Therefore, the method could obtain different phase responses of orthogonal polarization states simultaneously by changing the length and width of rectangular nano-bricks, and could independently manipulate the local phase of a pair of incident orthogonal polarization states.

## 3. Results and Discussions

### 3.1. Deviation of Height

To demonstrate, we started by investigating the influences of the first type of fabrication error, i.e., the deviation of height from the designed value, on focusing efficiency and FWHM. The focusing efficiency is defined as the ratio of light intensity at the focus to incident light intensity, and could be written as $\eta =I_{focus}/I_{in}$, where the *η* is focusing efficiency, *I_focus_* and *I_in_* are light intensity at the focus and incident light intensity, respectively. Figure 3a,b show the simulated distribution of the transmitted light intensity in the *x*–*z* plane under normal incidence with *x*- and *y*-polarization, respectively. The white solid lines plot the distributions of light intensity along the white dashed lines in each case. The position on the white dashed lines corresponded to the focal length of the ideal metalens, which could fully demonstrate the focusing performance of the metalens, including focusing efficiency and FWHM. Therefore, the light intensity distribution profiles at this location were facilitated to study the effect of fabrication errors on focusing performance. The numbers on the right side of each figure were the values of light intensity at the focus. From left to right, the heights of GaAs nano-bricks were *h* = 2.8 μm, 3.2 μm, 3.6 μm, 4.0 μm, and 4.4 μm, respectively. It was easy to see the focusing effect under both *x*- and *y*-polarization even though the fabrication error introduced a large difference in height (0.8 μm) between the real and ideal cases. Figure 3a shows that the focus intensity reduced with increasing deviation of the metasurface height compared with the ideal case under *x*-polarization. In contrast, the focus intensity under *y*-polarization did not show similar dependence on the height of GaAs nano-bricks, as shown in Figure 3b. As shown in Figure 3c, for the *x*-polarized incidence, the FWHM could be estimated as 0.71λ, 0.69λ, 0.83λ, 0.84λ, and 0.86λ when *h* = 2.8 μm, 3.2 μm, 3.6 μm, 4.0 μm, and 4.4 μm, respectively. For the *y*-polarized incidence, the FWHM could be estimated as 0.82λ, 0.81λ, 0.85λ, 0.88λ, and 0.92λ, when *h* = 2.8 μm, 3.2 μm, 3.6 μm, 4.0 μm, and 4.4 μm, respectively. The values of FWHM were not significantly distorted by the deviation of height for the cases of both *x*- and *y*-polarization, which showed the robustness of the focusing metasurface against this kind of fabrication error. We also summarized the focusing efficiency of the cases with different heights of GaAs nano-bricks under both *x*- and *y*-polarizations, as shown in Figure 3d. It was observed that the focusing efficiency under *x*-polarized incidence was smaller than that under *y*-polarized incidence, which was consistent with the results in Figure 3a,b. The difference in focusing performances between *x*- and *y*-polarization could be attributed to the asymmetric geometry of metalens along *x*- and *y*-axis in our simulation model.

### 3.2. Trapezoidal Error

The second type of fabrication error is the trapezoidal error, as shown in Figure 1c, which is also commonly introduced due to the unpredictable inclination. We set *h*_1_ = 1 μm and *h*_2_ = 2.6 μm. The fabrication error was modeled by changing the inclination angle *θ*. Figure 4a,b show the simulated distributions of the light intensity in *x*-*z* plane under normal incidences of *x*- and *y*-polarization, respectively. The white solid lines plot the distribution of light intensity along the white dashed lines in each case. The numbers on the right side of each figure are the values of light intensity at the focus. From left to right, we set *θ =* 4°, 8°, 12°, 16°, and 20°. Both Figure 4a,b show great focusing performance, even though the fabrication error of inclination angle was brought into the metasurface. The focusing electric field intensity did not weaken when the inclination angle increased under both *x*- and *y*-polarization, compared with the ideal case.

As shown in Figure 4c, for the *x*-polarized incidence, the FWHM could be estimated as 0.78λ, 0.71λ, 0.72λ, 0.71λ, and 0.72λ when *θ =* 4°, 8°, 12°, 16°, and 20°, respectively. For the *y*-polarized incidence, the FWHM could be estimated as 0.76λ, 0.76λ, 0.77λ, 0.81λ, and 0.83λ when *θ =* 4°, 8°, 12°, 16°, and 20°, respectively. The value of FWHM was not sensitive to the inclination angle for the cases of both *x*- and *y*-polarization. The results demonstrated that the focusing metasurface was robust against the inclination angles. Figure 4d summarizes the calculated focusing efficiency of each case, which was consistent with the results in Figure 4a,b. The difference in focusing performance between *x*- and *y*-polarization still existed in these cases.

### 3.3. Elliptical Error

We introduced the third type of geometric fabrication error. To investigate, the elliptical hemisphere height, as demonstrated in Figure 1d, was set as *h*_3_ = 0.2 μm, 0.4 μm, 0.6 μm, and 0.8 μm. Figure 5a,b show the simulated distributions of the light intensity in *x*-*z* plane under normal incidences of *x*- and *y*-polarization, respectively. The white solid lines plot the distributions of light intensity along the white dashed lines in each case. The numbers on the right side of each figure are the values of light intensity at the focus. From left to right, the results correspond to the cases with *h*_3_ = 0.2 μm, 0.4 μm, 0.6 μm, and 0.8 μm, respectively. Both Figure 5a,b show focusing performances of the metalens with the fabrication error of elliptical hemisphere. The focusing intensity became weaker in Figure 5a,b when the elliptical hemisphere height increased.

As shown in Figure 5c, for the *x*-polarized incidence, the FWHM could be estimated as 0.72λ, 0.73λ, 0.78λ, and 0.85λ when *h*_3_ = 0.2 μm, 0.4 μm, 0.6 μm, and 0.8 μm, respectively. For the *y*-polarized incidence, the FWHM could be estimated as 0.81λ, 0.85λ, 0.83λ, and 0.94λ when *h*_3_ = 0.2 μm, 0.4 μm, 0.6 μm, and 0.8 μm, respectively. The values of FWHM were not sensitive to the fabrication error of the elliptical hemisphere for the cases of both *x*- and *y*-polarization incidences. The results demonstrated that the focusing metasurface was also robust against the fabrication error of elliptical hemisphere. As summarized in Figure 5d, we found that the calculated focusing efficiencies were inconsistent with the simulated intensity distributions of the above focusing results. The existence of the fabrication error of the elliptical hemisphere might change the amplitude and phase of transmission, leading to deviation from the designed values. Through the simulation calculation, we could conclude that the focusing performance of the designed metasurface was quite robust against the fabrication error of elliptical hemisphere.

To better show the influences of the three fabrication errors, we plotted the simulated focusing efficiency of the metalens consisting of GaAs nano-bricks with different height (*h_1_*), inclination angle (*θ*) and elliptic hemisphere height (*h_3_*), as shown in Figure 6. It is easy to see that the focusing performance of the designed metasurface deviated from the desired values, due to the existence of fabrication errors. The focusing efficiencies under *x*-polarized incidence were different with those under *y*-polarized incidence. The difference could be attributed to the asymmetric geometry of the metalens along the *x*-axis and *y*-axis in our simulation model.

## 4. Conclusions

In summary, a focusing metasurface was designed, working at a wavelength of 4.2 μm, which consisted of GaAs nano-bricks standing on CaF_2_ substrate. The designed metasurface could focus the incident *x*- and *y*-polarized light at two symmetrical positions. We introduced three kinds of typical manufacturing defects of metasurfaces and investigated their influences on the focusing performances, such as the focusing efficiency and the FWHM. The simulated results showed that the focusing intensity and focusing efficiency changed due to the deviation of fabricated metasurface from the ideal case. Fortunately, the value of FWHM was insensitive to the fabrication errors. The analysis results may help in saving costs of experimental manufacturing and theoretical simulation.

## Figures and Tables

**Figure 1 micromachines-13-02098-f001:**
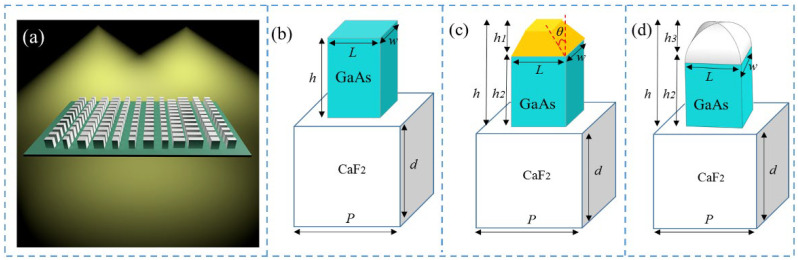
(**a**) Schematic of the metalens based on GaAs nano-bricks with different lengths and widths at the wavelength of 4.2 μm, with the incidence from substrate to free space. (**b**) Unit-cell of the proposed structure. The geometrical parameters were set as *P* = 1.8 μm, and *h* = 3.6 μm. Isosceles trapezoidal platform (**c**) and smooth elliptical hemisphere (**d**) on the top of GaAs nano-bricks, seaming completely with the rectangular brick.

**Figure 2 micromachines-13-02098-f002:**
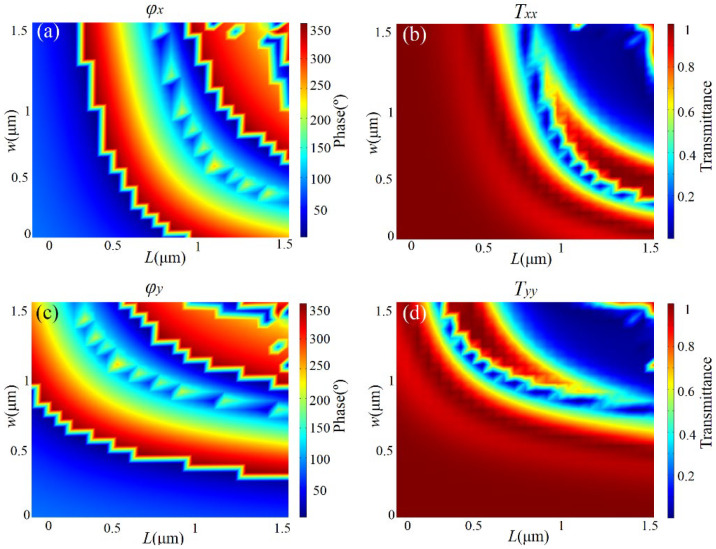
Transmitted phase (**a**) and normalized transmittance (**b**) as a function of *L* and *w* for normal incidence of the *X*-polarized light respectively, for the GaAs nano-bricks on CaF_2_ substrate. (**c**,**d**) Transmitted phase and normalized transmittance as a function of *L* and *w* for normal incidence of the *Y*-polarized light, respectively.

**Figure 3 micromachines-13-02098-f003:**
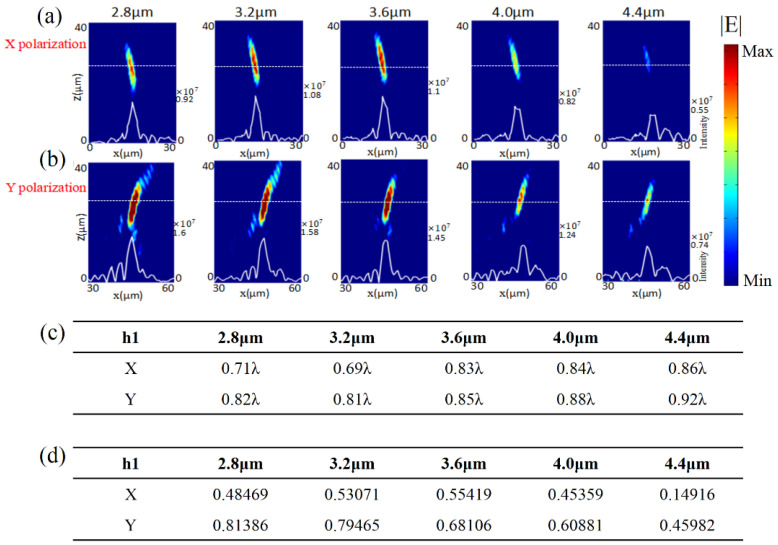
Focusing performance of the transmission metalens at a wavelength of 4.2 μm when the heights of GaAs nano-bricks were *h* = 2.8 μm, 3.2 μm, 3.6 μm, 4.0 μm, and 4.4 μm (from left to right). Distributions of light intensity in the *x*-*z* plane under normal incidences with (**a**) *x*- and (**b**) *y*-polarization. The white solid and dash lines are the intensity distribution curve and the position of focal plane, respectively. The number on the right side of each figure is the value of light intensity at the focus. (**c**) Simulated FWHM for each case. (**d**) Simulated focusing efficiency for each case.

**Figure 4 micromachines-13-02098-f004:**
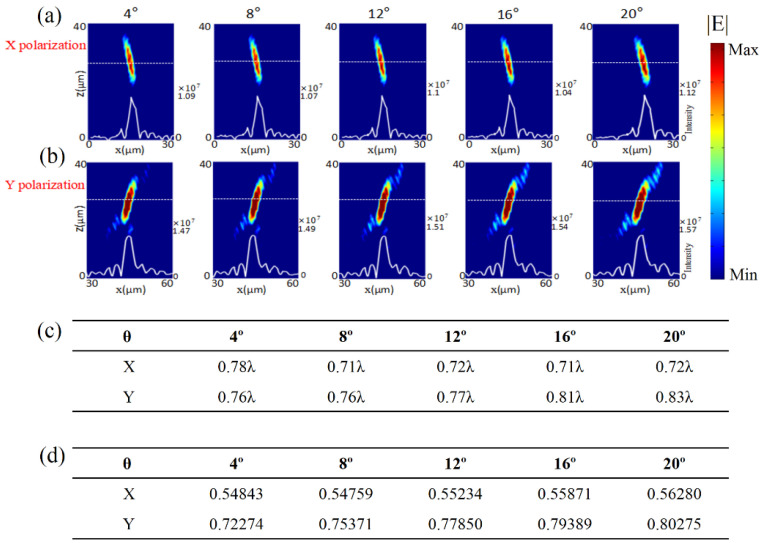
Focusing performances of the transmission metalens at a wavelength of 4.2 μm when inclination angles of *θ =* 4°, 8°, 12°, 16°, and 20° (from left to right) were introduced into the GaAs nano-bricks. Distributions of light intensity in the *x*-*z* plane under normal incidences of (**a**) *x*- and (**b**) *y*-polarization. The white solid and dash lines are the intensity distribution curve and the position of focal plane, respectively. The number on the right side of each figure is the value of light intensity at the focus. (**c**) Simulated FWHM for each case. (**d**) Simulated focusing efficiency for each case.

**Figure 5 micromachines-13-02098-f005:**
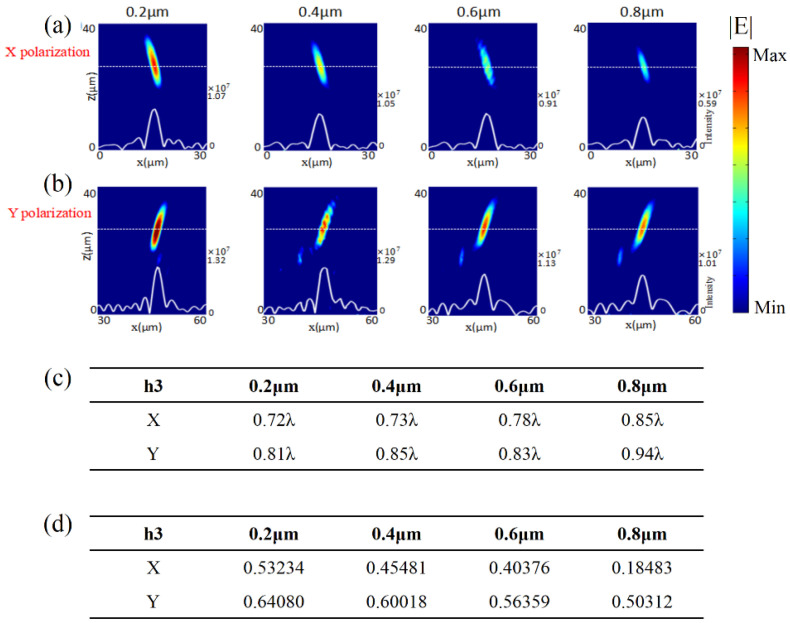
Focusing performances of the transmission metalens at a wavelength of 4.2 μm when the fabrication error of elliptical hemisphere, with *h*_3_ = 0.2 μm, 0.4 μm, 0.6 μm, and 0.8 μm (from left to right), emerged in the GaAs nano-bricks. Distributions of light intensity in the *x*-*z* plane under normal incidences of (**a**) *x*- and (**b**) *y*-polarization. The white solid and dash lines are the intensity-distribution curves and the positions of focal planes, respectively. The number on the right side of each figure is the value of light intensity at the focus. (**c**) Simulated FWHM for each case. (**d**) Simulated focusing efficiency for each case.

**Figure 6 micromachines-13-02098-f006:**
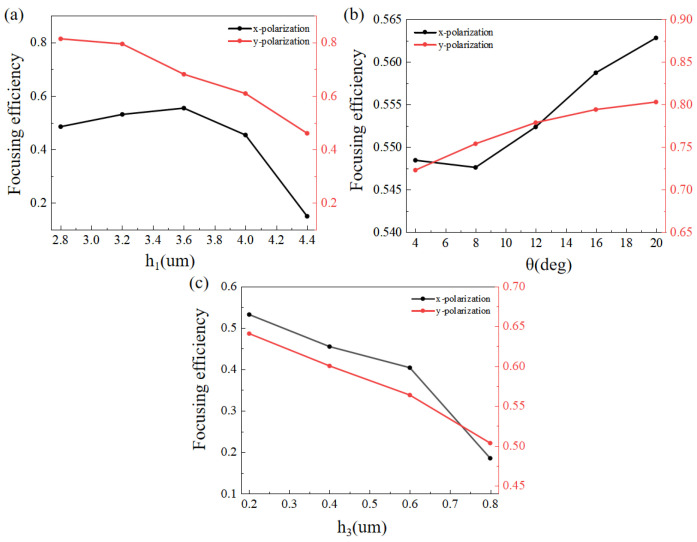
The Focusing efficiency of GaAs nano-bricks with (**a**) different height (*h_1_*), (**b**) inclination angle (*θ*) and (**c**) elliptic hemisphere height (*h_3_*) under *x*- and *y*-polarizations.

**Table 1 micromachines-13-02098-t001:** The structural parameters and corresponding phases of each nano-brick.

Element No.	Phase (deg.)	*L* (μm)	*w* (μm)	Element No.	Phase (deg.)	*L* (μm)	*w* (μm)
1	38	0.8	0.55	18	132	0.85	1.25
2	312	0.7	0.95	19	232	0.9	0.7
3	234	1.4	0.35	20	340	0.65	0.95
4	165	1.2	0.7	21	211	1	0.9
5	107	1.15	0.85	22	340	0.8	1.2
6	61	0.4	0.8	23	89	0.95	0.4
7	28	0.48	1.1	24	204	0.9	1.29
8	17	0.5	1.45	25	323	1	0.6
9	0	0.65	0.75	26	86	0.75	0.65
10	7	0.6	1	27	211	1.45	0.5
11	27	1.29	0.9	28	340	1.1	0.48
12	61	0.4	0.95	29	110	0.8	0.4
13	108	1.2	0.8	30	242	0.85	1.15
14	166	0.9	1	31	16	0.7	1.2
15	234	0.95	0.65	32	151	0.35	1.4
16	340	0.7	0.9	33	288	0.95	0.7
17	38	1.25	0.85	34	350	0.55	0.8

## Data Availability

Data available on request from the authors.

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
