# Peer review of "Influencing Effects of Fabrication Errors on Performances of the Dielectric Metalens"

_micromachines, 2022, doi:10.3390/mi13122098_

Round 1
Reviewer 1 Report
In this paper, the authors have proposed the investigation of the influences of different types of fabrication errors on the performance of focusing metalenses, such as the focusing efficiency and FWHM. However, the reviewer has some concerns.
1) The authors report a design of metalens and investigated the effect of 3 fabrication errors i.e. 1) the deviation of nanobrick height, (2) the trapezoidal error, and (3) the elliptical hemisphere height error. However, the work lacks innovation as there are already many papers demonstrating the influence of fabrication errors not only numerically but also experimentally, for instance, “Scientific Reports, 11, 5620 (2021)”, “IEEE Photonics Journal, Vol. 12, no. 3”, “ Computer Optics, Vol 42 (6), pp. 970-976”, “Micromachines (Basel). 2022 Sep 16;13(9):1531” and there is no comparison made with the existing work. So could the authors please elaborate on this?
2) The following paragraph seemed to be a misfit in the article so, upon investigation, it is realized that it belongs to another article (Nanomaterials 2021, 11(2), 260) of the authors and is an error in this article.
“Figure 1d shows the dependence of emissivity of the proposed thermal emitter on the crystalline fraction of GST under normal incidence. For the aGST MIM thermal emitter, four resonant emission peaks with average emissivity of 96% can be clearly seen in the simulated spectral emissivity in wavelength range of both 3–5 μm and 8–14 μm atmospheric windows, which are regarded as detected wavelength of IR detectors. These high emissivities do not satisfy the condition of thermal camouflage that requires radiant energy as little as possible in the middle-IR range (3–5 μm) and longer-IR range (8–14 μm). For the cGST MIM thermal emitter, only one broad resonant emission peak with emissivity of near unit at 6.3 μm can be seen, and meanwhile, the emissivity in the range of both 3–5 μm and 8–14 μm is relatively low, meeting the requirements of thermal camouflage. In addition, the near unit emissivity at 6.3 μm with FMHW of 2.7 μm can lead to the enhancement of radiation heat dissipation in the wavelength range 5–8 μm, thereby decreasing the temperature of objects and main- taining the thermal stability of system”.
3) Is there any specific reason for choosing the Isosceles trapezoidal platform and smooth elliptical hemisphere on the top of GaAs nanobricks.
4) No graphs of the simulated focusing efficiencies are added in the articles.
5) In its current state, the level of English throughout your manuscript does not meet the required standard. You may wish to ask a native speaker to check your manuscript for grammar, style, and syntax, or use the professional language editing options.

Reviewer 2 Report
In this manuscript, authors numerically demonstrate influences of fabrication errors in dielectric metalens on the focusing efficiency and the full width at half maximum (FWHM) values. The simulated results look scientifically sound. The innovation of this work lies in the introduction of common fabrication errors in micro-nano processing, demonstrating that the performance of dielectric metalens is robust to the fabrication errors within a certain range. Thus, I think this manuscript is worthy of peer review. However, I would like to the authors to address the following concerns to improve the manuscript.
1. In the fabrication process of metalens, which process will introduce the fabrication errors described in this paper? Please give detailed explanation and add relevant content to the manuscript.
2. To better show influences of fabrication errors in metalens on the full width at half maximum (FWHM) values, the authors are suggested to summarize the results in Figures. 3-5.
3. On page 5, line 15, the authors mentioned that "…It is easy to see the focusing effect under both x- and y-polarization even though the fabrication error introduced a large difference in height (1.2μm) between the real and ideal cases.", here, it can only be seen from the figure 3 that the maximum difference between the actual height and the ideal height is 0.8. Is the author's expression wrong, please correct it.
4. In addition, there are several typos in this manuscript. The reviewer suggests the authors to polish these sentences in the manuscript.
Reviewer 3 Report
Metalens are made by periodic planar arrays which can be designed to control the phase, amplitude, and polarization of the incident light. Metalens have a lot of potential applications in polarization transformation, optical focusing, imaging, etc. Therefore, it is important to study the nano-fabrication of Metalens and study how fabrication errors influence the performance of the Metalens. In this manuscript, the authors studied the influences of fabrication errors in dielectric metasurface-based metalens, including geometric sizes and shapes of the unit element, on the focusing efficiency and FWHM values. Their simulation results show the focusing intensity and focusing efficiency will change due to the deviation of the fabricated metasurface, while the value of FWHM is insensitive to fabrication errors. In my view, this manuscript is useful in the field of nano-optics. However, there are still some issues that must be addressed before I recommend it for publication in Micromachines.
1. In Figure 1, why the GaAs substrate has been highlighted by red lines?
2. Line 4-16, page 3, what is GST, this paragraph is non-sense and not related to this study. please consider rewriting or deleting it.
3. In all the figures and main text, the length L should be the same, either L or l.
4. In Figure 2, what are the color bars here, what are the units?
5. In Figures 3, 4, and 5. The format of the letter (a) (b) (c) should consistent with previous Figures.The authors should also include the color bar in these figures
6. Line 5-6, page 6, why did the authors choose this position to plot the intensity distribution profiles? The authors should elaborate.
Round 2
Reviewer 1 Report
Accept in present form
Reviewer 3 Report
The authors have satisfactorily responded to all my questions and made the necessary changes to the manuscript.